# Effects of Dietary Supplementation with Red Yeast (*Sporidiobolus pararoseus*) on Productive Performance, Egg Quality, and Duodenal Cell Proliferation of Laying Hens

**DOI:** 10.3390/ani12030238

**Published:** 2022-01-19

**Authors:** Chanidapha Kanmanee, Orranee Srinual, Montri Punyatong, Tossapol Moonmanee, Chompunut Lumsangkul, Suchon Tangtaweewipat, Hien Van Doan, Mongkol Yachai, Thanongsak Chaiyaso, Wanaporn Tapingkae

**Affiliations:** 1Department of Animal and Aquatic Sciences, Faculty of Agriculture, Chiang Mai University, Chiang Mai 50200, Thailand; chanidapha_k@cmu.ac.th (C.K.); orranee_sr@cmu.ac.th (O.S.); montri.pun@cmu.ac.th (M.P.); tossapol.m@cmu.ac.th (T.M.); chompunut.lum@cmu.ac.th (C.L.); suchon.t@cmu.ac.th (S.T.); hien.d@cmu.ac.th (H.V.D.); 2Innovative Agriculture Research Center, Faculty of Agriculture, Chiang Mai University, Chiang Mai 50200, Thailand; mongkol_yc@mju.ac.th; 3Faculty of Animal Science and Technology, Maejo University, Chiang Mai 50290, Thailand; 4Division of Biotechnology, Faculty of Agro-Industry, Chiang Mai University, Chiang Mai 50100, Thailand; thanongsak.c@cmu.ac.th

**Keywords:** antibiotic, yeast, performance, histology, laying hens

## Abstract

**Simple Summary:**

The present study investigated the effect of different levels of red yeast added to the diet of laying hens as a substitute for antibiotics. The aim of this study is to measure growth performance, egg quality, and small intestinal health of hens receiving this supplement at various levels during 22–60 weeks of age. The results indicate that supplementation with dietary red yeast has a positive effect on productivity and gut health; thus, we suggest administration of this additive as a substitute for antibiotics in laying hens.

**Abstract:**

Nowadays, industrial poultry producers are more focused on the safety of their products, especially contaminants from feedstuffs such as mycotoxin and pesticides. The residue from animal production using antibiotic growth promoters (AGPs) may cause some problems with antimicrobial resistance in human and animals. Red yeast (*Sporidiobolus pararoseus*) has a cell wall consisting of β-glucan and mannan-oligosaccharides and pigments from carotenoids that may be suitable for use as a substitute for AGPs. The objective was to evaluate the effects of red yeast in laying hen diets on productive performance, egg quality, and duodenal health. A total of 22-week-old laying hens (*n* = 480) were divided into five groups: control diet (CON), AGP at 4.5 g/kg and red yeast supplementation at 1.0 (RY1.0), 2.0 (RY2.0) and 4.0 g/kg (RY4.0) of diet. The results show that the AGP, RY2.0, and RY4.0 groups had significantly higher final body weight compared with the other groups (*p* < 0.001). The red yeast supplementation improved the egg shape index (*p* = 0.025), Haugh unit (*p* < 0.001), and yolk color (*p* = 0.037), and decreased yolk cholesterol (*p* < 0.001). Diet with red yeast supplementation improved villus height to crypt depth ratio and crypt cell proliferations. In conclusion, red yeast supplementation at 2.0 g/kg of diet can substitute AGP in layer diet.

## 1. Introduction

Raising laying hens must have adequate strategies for maintaining egg production [1]. Current advances in genetic selection, nutritional effectiveness, and health management for improvement of poultry production have resulted in better growth performance and greater longevity for poultry industry [2]. Antibiotic growth promoters (AGPs) can promote chicken performance by increasing feed efficiency and decreasing disease incidence [3]. However, due to the lack of awareness in some farmers and negligence of monitoring authorities, antibiotic residues in chicken meat were above the maximum permissible limits [4]. At this point, the outrageous use of AGPs in livestock is the important transmission pathway for antibiotic-resistant bacteria from animal products to humans, among which *Campylobacter* spp., *Salmonella* spp., and food-borne *Escherichia coli* are several common antibiotic-resistant pathogens [5]. Thus, it is always the important challenge of poultry producers to eliminate or reduce the use of AGPs in poultry production [6]. As alternatives to AGPs for poultry production, various bioactive compounds (e.g., prebiotics, antioxidants) have been widely investigated in laying hens [7].

Yeasts have the enzymes, vitamins, and other nutrients that have been shown to improve growth performance, feed efficiency, productive performance, reproduction, and internal egg quality in laying hens [8]. In laying hen production, yeast cell wall supplementations have been proven to enhance productive efficiency, egg quality, and egg yolk cholesterol contents [9,10]. In addition, dietary supplementation of yeast cell wall has been applied as a performance-enhancing alternative to in-feed antibiotics. Among these yeasts, red yeast (*Sporidiobolus pararoseus*) is a very promising one [11] since it naturally produces carotene pigment and cell wall functions as a prebiotic containing β-glucan and mannan-oligosaccharides (MOS). Carotenoids have been used in poultry feed as pigments in order to obtain the desired color of egg yolk or broiler meat and skin [12]. Addition of red yeast in laying hen diets was demonstrated to alleviate the beneficial impacts on feed efficiency, enhanced yolk color, and decreased cholesterol levels of serum and egg yolk as well as increased duodenal villus height [8,13].

Under the importance of intestinal structure and function in poultry, increased villus height is associated with increased absorptive surface and capacity of the intestines [14,15]. Moreover, a lower value of villus height (VH) to crypt depth (CD) ratio points to a lesser ability in nutrient digestibility and absorption in chickens [15,16]. Increased VH is associated with active cell mitosis (cell proliferation), which provides a higher absorptive potential of intestinal villi for nutrients [17,18]. In fact, a change of intestinal VH was generated by epithelial cell mitosis [19]. In poultry, mitotic index—as indicated by the ratio between the number of a population’s cells undergoing mitosis to its total number of cells—is able to indicate the duodenal cell proliferation and gut health [20].

All these observations indicate that supplementation with red yeast may positively affect laying performance and egg quality and lumen health in laying hens. Due to the importance of red yeast as prebiotic and natural colorant, enhanced knowledge of the development of red yeast supplementations as a substitute for antibiotics in laying hens is essential to increasing the application of red yeast in laying hens. However, as far as we know, there is limited information regarding the effects of red yeast on laying hen growth performance and small intestinal health. Therefore, the objective of this study was to evaluate the effect of dietary red yeast on productive performance, egg quality, and duodenal health (cell proliferation) of laying hens.

## 2. Materials and Methods

### 2.1. Animals and Experiment Design

The 22-week-old Hy-Line Brown hens (*n* = 480) were randomly divided into 5 dietary groups (96 hens/group). The experiment was a completely randomized design with 24 replications. The 5 dietary groups were: control diet (CON), control diet with AGP (Otamix a.c., Octa Memorial Co., Ltd., Bangkok, Thailand; consisted of amoxicillin 100 g/kg and colistin 400 × 10^6^ IU/kg at 4.5 g/kg), and control diet with red yeast supplementation at 1.0 g/kg (RY1.0), 2.0 g/kg (RY2.0) and 4.0 g/kg (RY4.0) of diet. Feed and clean drinking water were generally supplied and on ad libitum basis for 38 weeks. During the experimental period, the light/dark program was 16 h/8 h. The ingredients and nutrient values of the experimental diet are shown in Table 1. Red yeast was cultivated in a medium of yeast extract (4 g/L), malt extract (10 g/L), and glucose (4 g/L). An initial pH was adjusted to 6.0 and it was sterilized at 121 °C for 15 min [21]. After cultivation in the 5-L, 30-L, and 300-L bioreactors, the cultivated medium containing red yeast cell was stored at 4 °C in for 14 days to allow the autolysis and settle down of red yeast cell. The red yeast was mixed with corn starch at carrier 16% (w/v). Then, the mixture was subjected to drying process at 60 °C for 24 h to obtain the red yeast powder.

### 2.2. Productive Performance and Egg Quality Characteristics

Body weight of laying hens were assessed individually at the initiation and at the end of the experiment for calculation of growth performance. Dairy egg collection was expressed on a hen-day basis. On the first day of every week, eggs were collected and weighed individually. Daily feed intake was recorded and calculated as g day^−1^ hen^−1^. The feed conversion ratio was calculated by dividing the total feed consumption (g) by egg mass (g).

On days 1, 7, 14, 28, and 56, and every 28 days of the experimental period, 120 eggs laid between 08:00 and 12:00 h were randomly selected from each dietary group (3 eggs/replicate) to evaluate the characteristics of egg quality. The egg weight percentage was calculated as yolk and shell weights divided by whole egg weight. A digital egg tester DET6500 (NABEL Co., Ltd., Kyoto, Japan) was used to determine the egg shell thickness, egg shell strength, yolk color, and Haugh unit. A digital caliper was used to measure egg length and width and the egg shape index was calculated as egg width divided by egg length and multiplied 100. Egg shells index was calculated as shell weight divided by shell surface and multiplied by 100. The egg surface area was determined using the formula: Egg surface area = 4.67 × (egg weight)^2/3^. At the end of the experimental period, 24 eggs were randomly collected from each dietary group (3 eggs/replicate) and prepared based on a previous report [22] to determine egg yolk cholesterol. The cholesterol levels in egg yolk were evaluated by enzymatically using commercial reagent kits (Roche Diagnostic Systems Inc., Montclair, NJ, USA).

### 2.3. Histology of Duodenum

At the end of the experimental period (60 weeks of age), 24 hens per each group were euthanized and duodenal tissue samples were collected individually. Duodenal histology was evaluated according to a previous study [23]. Briefly, after euthanasia, duodenal tissues from each hen were harvested and subsequently immersed in a solution of phosphate-buffered saline (PBS; pH 7 at 4 °C). The duodenal tissue specimens were then fixed for 24 h with a 10% neutral-buffered formalin. After fixation, the tissue specimens underwent the tissue processing that consisted of 3-step process—dehydration, clearing, and impregnation with paraffin wax. Paraffin-embedded tissues were cut into 5 μm-thick sections (3 cross-sections/sample) using a rotary microtome and placed on glass slides. Deparaffinized tissue specimens were stained haematoxylin and eosin (H&E). After staining, duodenal tissue sections were covered with mounting medium and then covered with a glass coverslip. The H&E-stained histological slides were observed and visualized using a compound microscope (CX21, Olympus Cooperation, Tokyo, Japan) equipped with a digital video camera (Motic MC 2000), at 10× objective lens. The digital tissue images were examined using an image analyzer to measure villus height (VH; μm), villus width (VW; μm), crypt depth (CD; μm), crypt area (CA; μm^2^), and VH:CD ratio for each hen [24].

### 2.4. Duodenal Immunohistochemistry and Mitotic Index

To evaluate duodenal proliferations, immunolocalization of proliferating cell nuclear antigen (PCNA) in duodenal villi and crypt cells was used to visualize the S-phase cells in the cell cycle corresponding to the proliferation activity in the intestinal chicken mucosa. Staining for PCNA was adapted from previously described by Marchini et al. [25]. Briefly, deparaffinized sections were put in citrate buffer and subjected to microwave pretreatment for antigen retrieval. The tissue slides were placed inside the humidified chamber and treated with 3% hydrogen peroxide for 5 min to block endogenous peroxidase activity, followed by incubation with horse serum diluted 1:100 in PBS for 8 min. The tissue sections were then incubated for 15 min with diluted (1:300) primary antibody (mouse anti-PCNA, Leica Biosystems, Newcastle upon Tyne, UK). The slides were washed in wash buffer and incubated with secondary biotinylated anti-globulin G antibodies (Leica Biosystems, Newcastle upon Tyne, UK). After a new wash in wash buffer, the sensitivity was improved using the avidin-biotin technique (Leica Biosystems, Newcastle upon Tyne, UK) diluted 1:100 in PBS. The reaction was visualized by incubating the sections with 3,3-diaminobenzidine tetrahydrochloride (Leica Biosystems, Newcastle upon Tyne, UK). The slides were counterstained with haematoxylin for examination on a compound microscope. Then, the slides were made and the quantification of cells in proliferation activity was performed in the crypts’ region and along the villi of the duodenum. For control duodenal section, the PCNA antibody was replaced with normal mouse IgG (4 mg/mL). The control and positive duodenal sections for PCNA immunohistochemistry is illustrated in Figure 1.

Tissue section images of 10 crypts/sample and 10 villi/sample for duodenal cells with brown-staining nuclei (PCNA-positive cells) and blue-staining nuclei (PCNA-negative cells) were visualized from the crypts’ region and along the villus cells. The cell mitotic index was presented as the percentage of proliferating cells in the crypt and villus. The crypt cell mitotic index was calculated by dividing the total number of PCNA-positive nuclei by total crypt epithelial cells and multiplying by 100. The villus mitotic index was calculated by dividing the total number of PCNA-positive cells in a villus column by total number of cells in column and multiplying by 100.

### 2.5. Statistical Analysis

The collected data were analyzed using SPSS software (SPSS Inc., Chicago, IL, USA). The one-way analysis of variance (ANOVA) was used to determine the effects of red yeast supplementation on productive performance, egg quality, and mitotic index. The Duncan’s new multiple range test was used to differentiate the significance of treatment means among the dietary groups. Differences were considered significant at the *p*-value < 0.05 and tendencies at the 0.05 ≤ *p*-Value < 0.10.

## 3. Results

### 3.1. Growth Performance

Dietary red yeast supplementation did not have a significant (*p* > 0.05) effect on feed intake, egg weight, hen day production, egg mass, or feed conversion ratio of laying hens (Table 2). However, final body weight was significantly improved in AGP, RY2.0, and RY4.0 fed hens compared with the control and RY1.0 (*p* < 0.001).

### 3.2. Egg Quality

There were no significant differences (*p* > 0.05) in egg shell weight, surface area, egg shell index, egg shell strength, egg shell thickness, yolk weight, and albumen weight among dietary groups (Table 3). However, red yeast 2.0 g/kg fed hens were enhanced (*p* < 0.05) on egg shape index when compared with the control, RY0.5, and RY4.0 (*p* < 0.05). Haugh unit deceased in AGP fed hens compared with the others (*p* < 0.001). Moreover, RY1.0 and RY2.0 increased yolk color when compared with the control (*p* < 0.05). Yolk cholesterol decreased in RY2.0 and RY4.0 hens when compared with the control and AGP (*p* < 0.001).

### 3.3. Duodenal Histology

The duodenal histology of the H&E-stained duodenal sections of laying hens receiving the control diet (a), AGP (b), and RY1.0 (c), RY2.0 (d), and RY4.0 (e) diets is illustrated in Figure 2. The VH tended to be higher (*p* = 0.075) in laying hens receiving the RY4.0 diet compared with the control and AGP diets (5297.9 ± 254.8 μm vs. 4322.9 ± 258.5 μm and 4303.2 ± 219.6 μm); however, it was not different when compared with the RY1.0 (4558.9 ± 268.4 μm) and RY2.0 (5040.5 ± 272.3 μm) diets (Figure 3a). Supplementing laying hen diets with red yeast at 2.0 and 4.0 g/kg diet resulted in VW (1289.9 ± 60.7 μm and 1319.9 ± 57.2 μm) equivalent (*p* > 0.05) to that resulting from supplementation with AGP (1321.9 ± 51.5 μm; Figure 3b). The VW was lower (*p* < 0.05) in hens receiving the control and RY1.0 diets (1076.1 ± 68.4 μm and 1113.6 ± 64.6 μm) than in hens receiving the AGP, RY2.0, and RY4.0 (1321.9 ± 51.5 μm, 1289.9 ± 60.7 μm and 1319.9 ± 57.2 μm, respectively) diets (Figure 3b). The CD was greater (*p* < 0.05) in hens receiving the control and AGP diets (1411.6 ± 89.4 μm and 1348.8 ± 69.6 μm) than in hens receiving the RY1.0 (1079.2 ± 57.1 μm) diet; however, there was no difference in the CD among the AGP (1348.8 ± 69.6 μm), RY2.0 (1194.5 ± 66.4 μm), and RY4.0 (1171.8 ± 42.7 μm) diets (Figure 3c). The CA was greater (*p* < 0.05) in hens receiving the control and AGP diets (78,263.0 ± 7833.9 μm^2^ and 72,161.5 ± 5252.0 μm^2^) than in hens receiving the RY4.0 (52,123.8 ± 2919.8 μm^2^) diet; however, there was no difference in the CA among the AGP (78,263.0 ± 7833.9 μm^2^), RY1.0 (61,304.9 ± 5504.2 μm^2^), and RY2.0 (60,878.5 ± 5152.8 μm^2^) diets (Figure 3d). The VH:CD ratio was greater (*p* < 0.05) in hens receiving the RY1.0, RY2.0, and RY4.0 diets (4.8 ± 0.4, 4.9 ± 0.4, and 4.9 ± 0.3, respectively) than in hens receiving the control and AGP (3.5 ± 0.2 and 3.6 ± 0.3) diets (Figure 3e).

### 3.4. Duodenal Proliferations

The villus mitotic index was higher (*p* < 0.05) in laying hens receiving the RY2.0 diet (78.7 ± 2.1%) compared with the control and AGP diets (65.3 ± 3.1% and 70.9 ± 2.8%); however, it was not different when compared with the RY1.0 (74.5 ± 1.0%) and RY4.0 (77.0 ± 1.4%) diets (Figure 4a). The crypt cell mitotic index was greater (*p* < 0.05) for hens receiving the AGP, RY1.0, RY2.0, and RY4.0 (91.2 ± 1.6%, 94.0 ± 0.6%, 93.8 ± 1.0%, and 93.3 ± 0.7%) diets than for hens receiving the control (85.0 ± 2.5%) diet (Figure 4b).

## 4. Discussion

Although demand for animal protein is increasing, the use of antibiotics in animal feed has increased restrictions [26]. Especially, in laying hen feed, this restriction is probably due to the fact that the absorption efficiency of antibiotics was higher in laying hens than in broilers [27]. In addition, this results in antibiotic residues in animal-derived products and antibiotic resistance impacts related to public health concerns and food safety [27]. It has been proven that yeast supplementations improve productive performance, egg quality, and cholesterol levels of egg yolks of laying hens [9,10]; however, there is limited data about the inclusion of red yeast in the laying hen diet.

The addition of dietary red yeast in the diets of laying hens did not significantly affect feed intake, egg weight, hen day production, egg mass, or feed conversion ratio. Other researchers have found similar effects in poultry using yeast [28], yeast cell wall [9], yeast β-glucan [29], and prebiotic [30]. In contrast, the addition of inactive yeast [31], MOS [32], and β-carotene [33] to the diets of laying hens has been found to improve performance. In terms of final body weight, our findings revealed that it was improved by red yeast supplementation and antibiotics. Some researchers have previously indicated that feeding dried yeast [34], prebiotic [35], and β-carotene [33] to laying hens improved body weight. However, other researchers have found that the addition of yeast product did not affect body weight in laying hens [36]. Red yeast has been reported to demonstrate prebiotics properties—also defined as non-digestible feed components—that positively affect the welfare and health of the host by selectively stimulating the growth of beneficial bacteria or a limited population of harmful bacteria in the intestine [37].

There were no significant effects of dietary red yeast on egg shell weight, surface area, egg shell index, egg shell strength, egg shell thickness, yolk weight, or albumen weight. These results are consistent with the clam that laying hens fed yeast cell wall [9], yeast with bacteriocin [26], β-glucan [28], or prebiotic [30] had no effect on egg shell strength or yolk weight, whereas some effect on egg shell thickness had been observed. In the present study, it is interesting to note that dietary inclusion of red yeast improved egg shape index, as indicated the egg quality [38]. It has been reported previously that addition of selenium yeast in hen diets improved egg shape index in laying hens [39]. Whereas past researchers have found that supplementation of yeast and yeast cell wall improved Haugh unit [9,26], the present study has shown that addition of red yeast to hen diets improved Haugh unit, which is a major indicator of egg quality and freshness [40]. This implies that the supplementation of red yeast in hen diets can have beneficial effects on egg quality in laying hens. However, it has been reported previously that inclusion of dried yeast into rice husk-based diets for laying hens did not affect egg shape index [34]. As has been reported previously, the supplementation of dried yeast, β-glucan, and prebiotic has no effect on Haugh unit [29,30,34]. The different quality of eggs in response to yeast supplementation may have been due to the age of hens and yeast products and levels.

The higher index of egg yolk color observed in the red yeast treated group has been reported earlier by Tapingkae et al. [8]. However, the addition of yeast cell wall [9], yeast with bacteriocin [26], β-glucan [29], and prebiotic [30] had no effect on yolk color. Carotenoids have been included in poultry diets to improve the yellow pigmentation of broiler skin and meat as well as laying hen egg. The dietary supplementation of carotenoids to the poultry have also been demonstrated to increase the oxidative stability of poultry products. The antioxidant properties of carotenoids in food products may be responsible for their beneficial influence on human health [41]; for example, they offer remarkable synergic protection in the neurosensory retina indicated with enhanced risk reduction against age-related macular degeneration [42].

In laying hens fed red yeast, yolk cholesterol levels were lower. Other studies have found that supplementing the feed of laying hens with yeast or yeast products [31,35], grape seed extract with yeast culture [10], red yeast [8], and tomato waste meal containing carotenoid compounds [43] can help lower cholesterol levels in egg yolks. Chicken eggs are well-known for their high protein and vital nutritional content. However, the high cholesterol level of the yolk is a key limiting factor in its use in an effort to avoid blood cholesterol rise and minimize the risk of coronary heart disease [10]. The decrease in yolk cholesterol might be due to a decrease in cholesterol absorption or production in the gastrointestinal tract [10]. First, 3-hydroxy-3-methylgluteryl CoA (HMG-CoA) reductase catalyzes the production of mevalonate from HMG-CoA, in which the reaction of HMG-CoA is the rate-limiting step for cholesterol biosynthesis. Second, cholesterol in the body is eliminated primarily by converting it to bile acids. As a result, red yeast’s ability to lower cholesterol levels is important in the prevention of heart disease.

Interestingly, our findings highlight that the VH tended to be higher in laying hens receiving the red yeast supplementation, and that the VH:CD ratio was improved by red yeast supplementation. These results are consistent with the claim that the supplementation of *S. pararoseus* red yeast in laying hen diet [8] and *S. cerevisiae* yeast in broiler diet [44] improved gut health, as indicated by an increase in small intestinal histology. As stated above, a greater VH is indicated the active intestinal functions [45], which is paralleled by enhanced physiological functions of nutrient digestion and absorption in the intestine due to increase in absorptive surface area [46,47]. The higher digestion and absorption of nutrients are directly related to increased VH and VH:CD ratio that are associated with growth performance [48]. In the present study, it is interesting to note that a higher duodenal cell proliferation (villus and crypt cell mitotic indices) was observed in laying hens receiving the red yeast supplementation in diet. In fact, the primary components of yeast cell wall are β-glucan and MOS that are found in red yeast [13,49]. Supplementation of yeast β-glucan [50] and MOS [51] in diet can improve the intestinal histology of poultry. Moreover, MOS can promote beneficial bacteria, which improve gut health and boost host health [52]. As stated above, MOS may improve the fermentation of ingesta and leads to an increase in short-chain fatty acids (SCFAs) production [53]. In turn, SCFAs production by beneficial bacteria is a potentially essential effector of controlling cell proliferation in the small intestine [54] as an energy source for intestinal epithelium [55]. Although no evaluation of SCFAs production was attempted in the present experiment, our findings support the results of Penney et al. [56] who indicated that yeast cell wall hydrolyzes enhanced cell viability, which may result from increased proliferation or metabolic activity of the animal cells. This implies that the red yeast supplementation in laying hen diet benefited the gut health as indicated by increased duodenal histology and cell proliferations. On a cellular level, increased PCNA (protein which increases DNA polymerase activity) and mitotic index are described as indicators of cell proliferation [57]. Investigations in broilers receiving resistant starch (prebiotic property) have observed that duodenal mitotic index (the numerical percentage of PCNA-positive nuclei) was greater in chickens receiving a diet with 12% corn resistant starch than chickens receiving control diet [58]. Moreover, VH:CD ratio and mitotic index (the numerical percentage of PCNA-positive nuclei) in duodenum and ileum of small intestine were increased linearly in broilers fed with resistant starch diet [58]. This suggested that intestinal mitotic index might be a promising indicator for predicting cell proliferation in chickens.

## 5. Conclusions

Taken together, we highlight that the red yeast (2.0 and 4.0 g/kg of diet) and AGP supplementations in the laying hen diet had beneficial effects on final body weight. Dietary with red yeast supplementation also improved egg shape index, Haugh unit, and yolk color, decreased yolk cholesterol, and increased the duodenal mitotic index. Therefore, it can be concluded that red yeast supplementation at 2.0 g/kg of diet can be substituted for AGP in laying hen diet.

## Figures and Tables

**Figure 1 animals-12-00238-f001:**
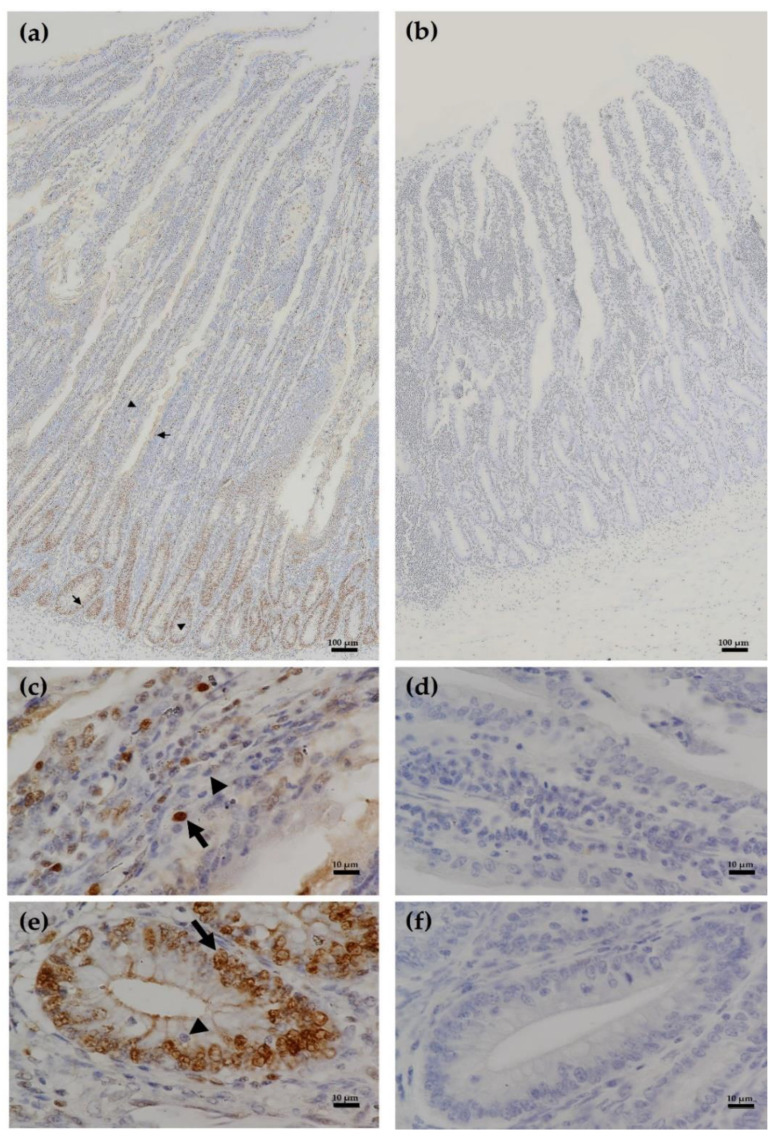
Immunoexpression of proliferating cell nuclear antigen (PCNA) in duodenal tissue sections of laying hens (**a**) for villi (**c**) and crypt cells (**e**) with brown-staining nuclei (PCNA-positive cells; arrows) and blue-staining nuclei (PCNA-negative cells; arrowheads). Control tissue sections (no primary antibody) did not exhibit any positive staining (**b**) in villi (**d**) and crypt cells (**f**). Magnifications were with ×10 (**a**,**b**) and ×100 (**c**–**f**) objective lens.

**Figure 2 animals-12-00238-f002:**
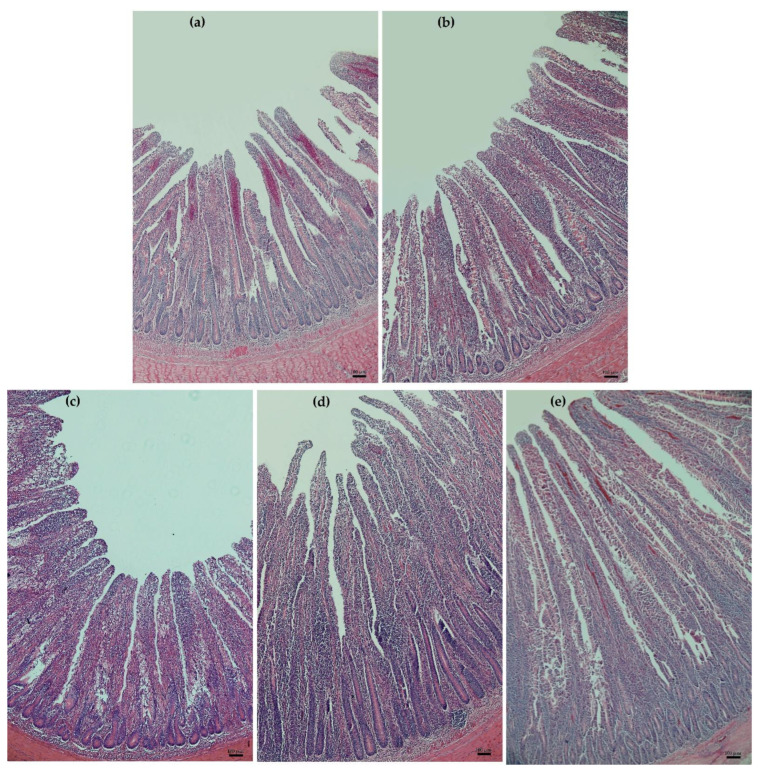
Histological representations of the H&E-stained duodenal sections of laying hens (*n* = 120 with 24 hens/group) receiving control diet (**a**), antibiotic growth promoter (**b**), and red yeast supplementation at 1.0 g/kg (**c**), 2.0 g/kg (**d**), and 4.0 g/kg (**e**) of diets. Magnification was with ×10 objective lens.

**Figure 3 animals-12-00238-f003:**
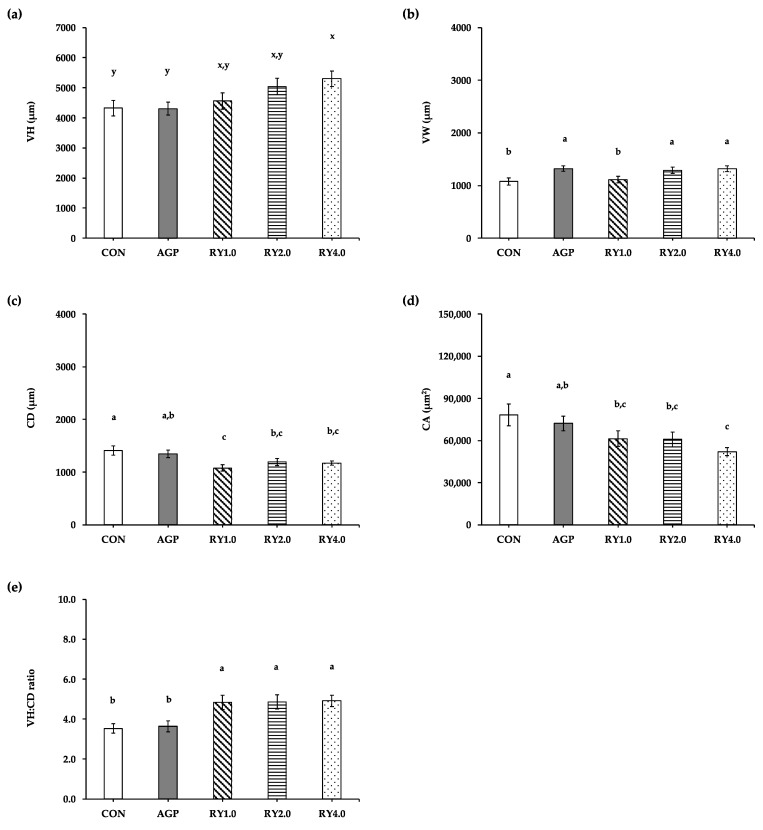
Means (±SEM) of the VH (**a**), VW (**b**), CD (**c**), CA (**d**), and VH:CD ratio (**e**) in laying hens (*n* = 120 with 24 hens/group) receiving control diet (CON), antibiotics (AGP; amoxicillin and colistin at 4.5 g/kg), and red yeast supplementation at 1.0 g/kg (RY1.0), 2.0 g/kg (RY2.0), and 4.0 g/kg (RY4.0). x,y Values with different superscript letters tend to differ between groups at 0.05 ≤ *p*-value < 0.10. a,b,c Values with different superscript letters indicate significant differences among groups at *p*-value < 0.05. VH, villus height; VW, villus width; CD, crypt depth; CA, crypt area; VH:CD ratio, villus height with crypt depth ratio.

**Figure 4 animals-12-00238-f004:**
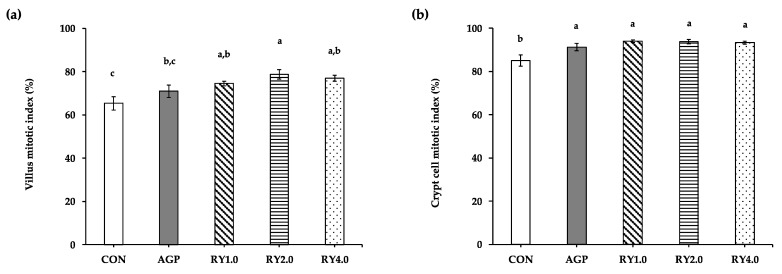
The means (±SEM) of the villus (**a**) and crypt cell (**b**) mitotic indicus in laying hens (*n* = 120 with 24 hens/group) receiving control diets (CON), antibiotic (AGP; amoxicillin and colistin at 4.5 g/kg), and red yeast supplementation at 1.0 g/kg (RY1.0), 2.0 g/kg (RY2.0), and 4.0 g/kg (RY4.0). a,b,c Values with different superscript letters indicate significant differences among groups at *p*-value < 0.05.

**Table 1 animals-12-00238-t001:** Ingredients and chemical composition of the experimental diets.

Item	g/kg
Ingredients	
Rice bran	60.0
Corn meal	534.0
Soy bean meal ^1^	192.0
Fish meal ^2^	74.0
Leucaena leaf meal	20.0
Shell flour	81.0
Dicalcium phosphate	6.0
Sodium chloride	5.0
Vegetable oil	25.0
DL-methionine	0.5
Premix ^3^	2.5
Nutrient value	
Calculated analysis	
Metabolizable energy (kcal/kg)	2753.14
Calcium	38.7
Phosphorus	5.4
Sodium	3.7
Choline	4.6
Lysine	9.9
Methionine	3.7
Methionine and cysteine	6.5
Tryptophan	2.1
Linoleic	17.0
Fat	67.1
Crude fiber	38.1
Proximate analysis	
Dry matter	895.4
Ash	108.9
Crude fiber	41.1
Ether extract	40.1
Crude protein	208.3

^1^ 44% crude protein. ^2^ 55% crude protein. ^3^ Supplied vitamin A (12,000,000 IU/kg diet), vitamin D3 (2,400,000 IU/kg diet), vitamin E (30 g/kg diet), vitamin K3 (2.5 g/kg diet), vitamin B1 (2.5 g/kg diet), vitamin B2 (6 g/kg diet), vitamin B6 (4 g/kg diet), vitamin B12 (20 mg/kg diet), niacin (25 g/kg diet), calcium D-pantothenate (8 g/kg diet), folic acid (1 g/kg diet), vitamin C (50 g/kg diet), D-biotin (50 mg/kg diet), choline chloride (150 g/kg diet), canthaxanthin (1.5 g/kg diet), apo-carotenoic acid ester (0.5 g/kg diet), manganese (80 g/kg diet), zinc (60 g/kg diet), iron (60 g/kg diet), copper (5 g/kg diet), iodine (1 g/kg diet), cobalt (0.5 g/kg diet), and selenium (0.15 g/kg diet).

**Table 2 animals-12-00238-t002:** The growth performance and productivity in laying hens receiving (*n* = 480 with 96 hens/group) control diet (CON), antibiotic growth promoter (AGP), and red yeast supplementation at 1.0 g/kg (RY1.0), 2.0 g/kg (RY2.0), and 4.0 g/kg (RY4.0) of diets.

Item	CON	AGP ^1^	RY1.0	RY2.0	RY4.0	SEM	*p*-Value
Initial body weight (g)	1814.27	1811.56	1813.54	1813.23	1815.31	1.18	0.906
Final body weight (g)	1909.36 ^b^	1980.73 ^a^	1868.53 ^b^	1986.51 ^a^	1974.89 ^a^	9.85	<0.001
Feed intake (g/hen/day)	110.27	107.47	109.69	110.95	110.73	0.66	0.475
Egg weight (g)	61.94	60.83	62.07	62.60	61.60	0.22	0.128
Hen day production (%)	88.75	90.79	88.09	90.07	89.75	0.45	0.354
Egg mass	54.86	55.14	54.61	56.39	55.24	0.31	0.436
Feed conversion ratio	2.02	1.96	2.02	1.98	2.01	0.01	0.479

^1^ Antibiotic (amoxicillin and colistin at 4.5 g/kg); RY 1.0, red yeast supplementation at 1.0 g/kg; RY 2.0, red yeast supplementation at 2.0 g/kg; RY 4.0, red yeast supplementation at 4.0 g/kg. ^a,b^ Mean values with different letters in the same row indicate significant differences (*p*-value < 0.05). SEM, standard error of measurement.

**Table 3 animals-12-00238-t003:** The egg quality and yolk cholesterol in laying hens (*n* = 120 with 24 hens/group) receiving control diet (CON), antibiotic growth promoter (AGP), and red yeast supplementation at 1.0 g/kg (RY1.0), 2.0 g/kg (RY2.0), and 4.0 g/kg (RY4.0) of diets.

Item	CON	AGP ^1^	RY1.0	RY2.0	RY4.0	SEM	*p*-Value
Egg shape index (%)	77.38 ^b^	77.62 ^a,b^	77.56 ^b^	78.27 ^a^	77.20 ^b^	0.11	0.025
Shell weight percentage (%)	14.35	14.14	14.17	13.85	14.30	0.07	0.232
Surface area (m^2^)	71.21	70.96	71.39	72.34	71.39	0.17	0.121
Egg shell index (%)	12.10	11.86	11.94	11.71	11.99	0.06	0.276
Egg shell strength (kgf)	4.40	4.32	4.30	4.28	4.27	0.03	0.518
Egg shell thickness (mm)	0.47	0.47	0.48	0.47	0.48	0.00	0.202
Yolk weight percentage (%)	24.57	24.34	24.48	24.12	24.63	0.08	0.281
Albumen weight percentage (%)	61.09	61.05	61.35	62.03	61.07	0.13	0.167
Haugh unit	91.52 ^a^	87.90 ^b^	90.59 ^a^	91.82 ^a^	90.90 ^a^	0.32	<0.001
Yolk color	8.99 ^b^	9.08 ^a,b^	9.18 ^a^	9.22 ^a^	9.13 ^a,b^	0.02	0.037
Yolk cholesterol (mg/g)	18.71 ^a^	18.14 ^a^	17.17 ^b^	15.76 ^c^	15.37 ^c^	0.25	<0.001

^1^ Antibiotic (amoxicillin and colistin at 4.5 g/kg); RY 1.0, red yeast supplementation at 1.0 g/kg; RY 2.0, red yeast supplementation at 2.0 g/kg; RY 4.0, red yeast supplementation at 4.0 g/kg. ^a,b,c^ Mean values with different letters in the same row indicate significant differences (*p*-value <0.05). SEM, standard error of measurement.

## Data Availability

Data available on request due to restrictions, e.g., privacy or ethical. The data presented in this study are available on request from the corresponding author. The data are not publicly available due to the law of the Ministry of Higher Education, Science, Research and Innovation.

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
