# Peer review of "Effects of Dietary Supplementation with Red Yeast (Sporidiobolus pararoseus) on Productive Performance, Egg Quality, and Duodenal Cell Proliferation of Laying Hens"

_animals, 2022, doi:10.3390/ani12030238_

Round 1
Reviewer 1 Report
The paper entitled "Effects of dietary red yeast (Sporidiobolus pararoseus) on layer performance, egg quality and duodenal cell proliferation" aims to explore whether red yeast can replace AGP in layer diets. It was found that the supplement of red yeast improved egg shape index, Haugh units and egg yolk color, and lowered egg yolk cholesterol, improved villus height, crypt depth ratio and crypt cell proliferation. The results of this article are sufficient, and the methods are detailed, but the writing is messy and there are places that are not careful enough. The following questions and suggestions are put forward for this research.
Abstract:
- In the abstract of Line31, is GAP a typo? It should be AGP?
- The summary does not reflect how many days the entire experiment lasted.
Introduction:
- There is no description of the harmfulness of AGP in the first paragraph of the introduction, and no relevant literature support. Please use the results of previous studies to illustrate the current research status of AGP antimicrobial resistance and antibiotic residues.
- The second paragraph of the introduction mentions "red yeast is a carotene pigment" (Line 57 and Line 59) twice, which I think is a bit long-winded. The concept and composition or source of red yeast should be summarized in one sentence each.
- Write the full name of red yeast for the first time, and you can omit the Latin word "Sporidiobolus pararoseus" afterwards.
- The second paragraph of the introduction does not reflect the beneficial effects of adding red yeast. Please add red yeast's research on other animals or different species.
- Please state the importance and significance of this article at the end of the introduction.
Materials and Methods:
- How to choose the amount of AGP and red yeast in the experiment?
- Please provide the manufacturer, purity and activity of AGP and red yeast.
- Why detect the mucosal proliferation activity of the duodenum of laying hens? Does this have anything to do with the quality of the eggs?
- Line 97, you said that each group made 24 chicken wax blocks. Can you upload all the HE staining pictures in the supplementary materials?
Results:
- Line 160, the Table 3 mentioned here does not match the Line 213-214 presented. Table 3 is not found.
- Most of the results did not mention how many replicates were performed for each experiment, please add "n=?" in the legend.
- Which two groups are the significant differences indicated by the p-values in Table 2 and Table 3?
Discussion:
- Line 286, “Carotenoids in the human food chain be advantageous for human” has a grammatical error, please correct it.
- Line 316, the font size of "yeast" is different from others, please unify.
Author Response
January 08, 2022
Assistant Editor: Animals
Email: charlotte.shen@mdpi.com
Dear Editor of Reviewer 1,
Thank you for the review report of editor for manuscript (ID. Animals-19-1531926) included information that the manuscript needs to revise according to the comment of the reviewers. We have taken the useful comments of the referees and editor into account, and have made major revision as following page. We also describe all changes using the Track Changes and mostly agree with the comments. Thus, we hope our ability of the revised version will be satisfied by the reviewers and editor.
After the major revision, we are fully aware of the revised submission and respectfully encourage you to give your utmost consideration to this revision manuscript. If you have any questions concerning with the manuscript, please feel free to contact me.
Sincerely yours,
Wanaporn Tapingkae, Ph.D
Corresponding author
e-mail: wanaporn.t@cmu.ac.th
In reply to review report of Assistant Editor dated December 29, 2021.
Subject: [Animals] Manuscript ID: animals-1531926 – Major Revisions (Due Date: 9 January 2022)
We have taken the useful comments and queries of the referees and editor into account, and have made major revision as follows (answers are in the italic font).
Reviewer 1:
The paper entitled "Effects of dietary red yeast (Sporidiobolus pararoseus) on layer performance, egg quality and duodenal cell proliferation" aims to explore whether red yeast can replace AGP in layer diets. It was found that the supplement of red yeast improved egg shape index, Haugh units and egg yolk color, and lowered egg yolk cholesterol, improved villus height, crypt depth ratio and crypt cell proliferation. The results of this article are sufficient, and the methods are detailed, but the writing is messy and there are places that are not careful enough. The following questions and suggestions are put forward for this research.
Comments and Suggestions for Authors:
Abstract:
- In the abstract of Line31, is GAP a typo? It should be AGP?
- Due to the typing error, thus, we did correct the word ‘GAP’ to ‘AGP’ as shown (Line 32 of revised manuscript with Track Changes).
- The summary does not reflect how many days the entire experiment lasted.
- We did agree with the suggestion of the reviewer. Thus, we did add the numbers of day in experiment as shown (Line 20 of revised manuscript with Track Changes).
Introduction:
- There is no description of the harmfulness of AGP in the first paragraph of the introduction, and no relevant literature support. Please use the results of previous studies to illustrate the current research status of AGP antimicrobial resistance and antibiotic residues.
- We definitely agree with the reviewer. Thus, we did improve the contents by adding the information of the current research status of antibiotic residues in chicken meat and the important cause of resistance transmission to humans through animal products as shown (Lines 48-54 of revised manuscript with Track Changes).
- The second paragraph of the introduction mentions "red yeast is a carotene pigment" (Line 57 and Line 59) twice, which I think is a bit long-winded. The concept and composition or source of red yeast should be summarized in one sentence each.
- We definitely agree with the reviewer. Thus, we did make the new sentence as follows.
- Among these yeasts, red yeast (Sporidiobolus pararoseus) is a very promising one since its naturally produces carotene pigment and cell wall functions as a prebiotic containing β-glucan and mannan-oligosaccharides (MOS) (Lines 65-67 of revised manuscript with Track Changes).
- Write the full name of red yeast for the first time, and you can omit the Latin word "Sporidiobolus pararoseus" afterwards.
- We did agree with the suggestion of the reviewer. The Latin name of "Sporidiobolus pararoseus" were omitted after the full name of red yeast was firstly addressed.
- The second paragraph of the introduction does not reflect the beneficial effects of adding red yeast. Please add red yeast's research on other animals or different species.
- We did agree with the reviewer. Thus, we did improve the contents by adding the information of red yeast's research on other animals or different species as shown (Lines 70-72 of revised manuscript with Track Changes).
- Please state the importance and significance of this article at the end of the introduction.
- We definitely agree with the reviewer. Thus, we did make the new sentences to state the importance and significance of this article at the end of the introduction as shown (Lines 82-86 of revised manuscript with Track Changes).
Materials and Methods:
- How to choose the amount of AGP and red yeast in the experiment?
- We did choose the amount of AGP from dose amoxicillin 20 mg/kg body weight/day and colistin 100,000 IU/kg weight/day and concentrate of antibiotic at amoxicillin 100 g/kg and colistin 400×106 IU/kg on body weight animals by feed intake 100 g/bird/day. The calculated of antibiotic at 4.5 g/kg of diet. In this study, the red yeast product was prepared by drying the red yeast cell at 60°C with using corn starch at carrier while the red yeast product of study was spray dried red yeast (Tapingkae et al., 2018). Tapingkae et al. (2018) demonstrated that supplementation with red yeast (S. pararoseus) at levels of 2.0 g/kg in laying hen diet. So, we choose the amount of red yeast in the experiment at 1.0, 2.0, and 4.0 g/kg of diet.
- Tapingkae, W.; Panyachai, K.; Yachai, M.; Doan, H.V. Effects of Dietary Red Yeast (Sporidiobolus pararoseus) on Production Performance and Egg quality of Laying Hens. J. Anim. Physiol. Anim. Nutr. 2018, 102, e337-e344.
- Please provide the manufacturer, purity and activity of AGP and red yeast.
- The manufacturer and purity of AGP was added (Lines 98-99 of revised manuscript with Track Changes).
- Besides, the details of red yeast were also added (Lines 103-109 of revised manuscript with Track Changes).
10. Why detect the mucosal proliferation activity of the duodenum of laying hens? Does this have anything to do with the quality of the eggs?
- The duodenum is very active in digestion and absorption processes as well as lower goblet cells density and mucus secretion may enhance absorptive capacity. Moreover, goblet cells in the duodenum were shown to store avidin, lysozyme and other secretory components. Secretory components have neutralising properties against pathogen-associated molecules and act as antibacterial substances. Thus, we did interest to evaluate mucosal proliferation activity of the duodenal health of laying hens.
11. Line 97, you said that each group made 24 chicken wax blocks. Can you upload all the HE staining pictures in the supplementary materials?
- Yes, we did provide the HE staining pictures in the Figure 2 as shown.
Results:
12. Line 160, the Table 3 mentioned here does not match the Line 213-214 presented. Table 3 is not found.
- Due to the typing error, thus, we did correct the number of table as shown (Line 267 of revised manuscript with Track Changes).
13. Most of the results did not mention how many replicates were performed for each experiment, please add "n=?" in the legend.
- We did agree with the reviewer. Thus, we did add the number of laying hens in the legends of all tables (Lines 262 and 267 of revised manuscript with Track Changes) and figures (Lines 300, 305 and 312 of revised manuscript with Track Changes) as shown.
- Which two groups are the significant differences indicated by the p-values in Table 2 and Table 3?
- The explanation of results in Tables 2 and 3 was rewritten to indicate the difference of treatment groups (Lines 189-190 and Lines 194-199 of revised manuscript with Track Changes).
Discussion:
15. Line 286, “Carotenoids in the human food chain be advantageous for human” has a grammatical error, please correct it.
- We did agree with the reviewer. Thus, we did correct grammatical errors as shown (Lines 380-381 of revised manuscript with Track Changes).
16. Line 316, the font size of "yeast" is different from others, please unify.
- We did correct the font size as shown (Line 411 of revised manuscript with Track Changes).

Reviewer 2 Report
The manuscript titled ‘Effects of Dietary Red Yeast (Sporidiobolus pararoseus) on Productive Performance, Egg Quality, and Duodenal Cell Proliferation of Laying Hens’ has been drafted fairly. The research concept is good. However, the authors need to revise their manuscript to improve its quality before it being accepted for publication. The general and specific comments have been provided as follows:
General comments
- What is the source of red yeast? Provide its composition as much as possible. What was its form (powder, flakes) used in the feed mix? The additives are in g/kg while the feed composition (table 1) is in percentage. Check the Animals guidelines for the SI unit and make them consistent.
- How was feed formulated? Were the supplements/additives added on top or by replacing any ingredients?
- The abstract should show more results with p values.
- Also, briefly explain the egg-related parameters in materials and method even if the references have been provided. Concise information on the Haugh unit, yolk color, the eggshell index would be required for a smooth flow in reading.
- Tell us more about the mitotic index in the introduction. What is its importance in relation to increased villus height and reduced crypt depth ratio? Discussion on mitotic index and cell proliferation is limited. It is an important finding of this study and should be discussed in detail to provide a better perspective.
- Line 263, 278, and elsewhere in the discussion section: The authors need to present the outcomes differently in discussion to avoid it being a repetition of what has been mentioned in the results. When the results and discussion is separate sections, the individual effect, or the collection of outcomes from results should not be copied in a similar tone in discussion. The rewriting of results looks like the topic. Please see below if the explanation is clear to the authors and would help improve their manuscript.
As a suggestion: the author could present it in a more expressive style linking with what has been observed in the results for immersive reading. It is hard to say about writing styles as these are individual-specific; however, it could be refined to present results in discussion from the point of view of the researcher. If it has been stated in the result that some parameters have increased, the authors would present in the discussion such as;
- the improvement noted in … in terms of … signifies .. and has been reported previously or is a unique finding not observed earlier..
eg. The higher index of egg yolk color observed in the red yeast treated group has been reported earlier by ..
Specific comments
Line 28–29: Rephrase this sentence.
Line 31: Correct ‘GAP’.
Line 34: Villus height ‘to’ crypt depth.
Line 36: Remove ‘for’ .
Line 90: Change ‘wide’ to ‘width’.
Line 146: Check for typo ‘≤ vs ≥’.
Line 158–160: Rephrase this sentence.
Line 162–163: Change ‘compared to’ to ‘compared with’.
Table 1: Check if ‘tryptophan’ would be better. Also, change fiber to ‘crude fiber’.
Line 236–238: Rephrase this sentence and provide a citation.
Line 248–249: Change ‘term’ to ‘terms’.
Line 269–270: Rephrase this sentence. Consistent would be with findings/results.
Line 286–288: Check this sentence.
Line 301–303: Why is red yeast an alternative source of antibiotics? Also, what does cholesterol do for antibiotic effect? This sentence would be an overstatement.
Line 312: Better to say ‘absorption of nutrients’. Substances could be nutritious or toxic.
Line 314: Change ‘indicus’ to ‘indices’.
Line 318–320: Rephrase this sentence. Improved gut health is the outcome of other effects.
Author Response
January 08, 2022
Assistant Editor: Animals
Email: charlotte.shen@mdpi.com
Dear Editor and Reviewer 2,
Thank you for the review report of editor for manuscript (ID. Animals-19-1531926) included information that the manuscript needs to revise according to the comment of the reviewers. We have taken the useful comments of the referees and editor into account, and have made major revision as following page. We also describe all changes using the Track Changes and mostly agree with the comments. Thus, we hope our ability of the revised version will be satisfied by the reviewers and editor.
After the major revision, we are fully aware of the revised submission and respectfully encourage you to give your utmost consideration to this revision manuscript. If you have any questions concerning with the manuscript, please feel free to contact me.
Sincerely yours,
Wanaporn Tapingkae, Ph.D
Corresponding author
e-mail: wanaporn.t@cmu.ac.th
In reply to review report of Assistant Editor dated December 29, 2021.
Subject: [Animals] Manuscript ID: animals-1531926 – Major Revisions (Due Date: 9 January 2022)
We have taken the useful comments and queries of the referees and editor into account, and have made major revision as follows (answers are in the italic font).
Reviewer 2:
General comments:
1. What is the source of red yeast? Provide its composition as much as possible. What was its form (powder, flakes) used in the feed mix? The additives are in g/kg while the feed composition (table 1) is in percentage. Check the Animals guidelines for the SI unit and make them consistent.
- We did add the information of red yeast including source, cultivation method, form, and compositions as shown (Lines 98-99 and 103-109 of revised manuscript with Track Changes).
- We did correct the unit in Table 1 to g/kg as shown.
2. How was feed formulated? Were the supplements/additives added on top or by replacing any ingredients?
- The red yeast was added on top to the control diet.
3. The abstract should show more results with p values.
- We did agree with the reviewer. Thus, we did add the p-values as shown (Lines 33-35 of revised manuscript with Track Changes).
4. Also, briefly explain the egg-related parameters in materials and method even if the references have been provided. Concise information on the Haugh unit, yolk color, the eggshell index would be required for a smooth flow in reading.
- We did agree with the reviewer. Thus, we did revise the contents by adding the explanation of materials and method in egg quality parameters as shown (Lines 120-121 and 125-128 of revised manuscript with Track Changes).
5. Tell us more about the mitotic index in the introduction. What is its importance in relation to increased villus height and reduced crypt depth ratio? Discussion on mitotic index and cell proliferation is limited. It is an important finding of this study and should be discussed in detail to provide a better perspective.
- We did agree with the reviewer. Thus, we did revise the contents by adding the information of mitotic index in the introduction section as follows.
- In poultry, mitotic index, as indicated by the ratio between the number of a population’s cells undergoing mitosis to its total number of cells, is able to indicate the duodenal cell proliferation and gut health [20] (Lines 80-82 of revised manuscript with Track Changes).
- We did revise the contents by adding the information of importance in relation to increased villus height and reduced crypt depth ratio as follows.
- Under the importance of intestinal structure and function in poultry, increased villus height is associated with increased absorptive surface and capacity of the intestines [14,15]. Moreover, a lower ratio of villus height and crypt depth indicates to a smaller capacity in nutrient digestibility and absorption in poultry [15,16]. Increased villus height is related with active cell mitosis (cell proliferation), which provides a greater absorptive potential of villus cells for nutrients [17,18]. In fact, a change of intestinal villus height was generated by epithelial cell mitosis [19] (Lines 74-80 of revised manuscript with Track Changes).
- Moreover, we did revise the contents by adding the information of mitotic index and cell proliferation in the discussion section as follows.
- On a cellular level, increased PCNA (protein which increases DNA polymerase activity) and mitotic index are described as indicators of cell proliferation [57]. Investigations in broilers receiving resistant starch (prebiotic property) have observed that duodenal mitotic index (the number of PCNA-positive cells divided by the total number of cells counted) was greater in chickens receiving diet with 12% corn resistant starch than chickens receiving control diet [58]. Moreover, VH:CD ratio and mitotic index (the percentage of PCNA-positive cells) in duodenum and ileum were increased linearly by feeding broilers with resistant starch diet [58]. This suggested that intestinal mitotic index might be a promising indicator for predicting cell proliferation in chickens (Lines 424-433 of revised manuscript with Track Changes).
6. Line 263, 278, and elsewhere in the discussion section: The authors need to present the outcomes differently in discussion to avoid it being a repetition of what has been mentioned in the results. When the results and discussion is separate sections, the individual effect, or the collection of outcomes from results should not be copied in a similar tone in discussion. The rewriting of results looks like the topic. Please see below if the explanation is clear to the authors and would help improve their manuscript.
As a suggestion: the author could present it in a more expressive style linking with what has been observed in the results for immersive reading. It is hard to say about writing styles as these are individual-specific; however, it could be refined to present results in discussion from the point of view of the researcher. If it has been stated in the result that some parameters have increased, the authors would present in the discussion such as;
the improvement noted in … in terms of … signifies .. and has been reported previously or is a unique finding not observed earlier..
The higher index of egg yolk color observed in the red yeast treated group has been reported earlier by ..
- We definitely agree with the reviewer. Thus, we did improve the contents by modifying the sentences and make the new sentences as shown (Lines 344-356 and 372-375 of revised manuscript with Track Changes).
Specific comments
Line 28–29: Rephrase this sentence.
- We did improve the contents by modifying the sentences and make the new sentences as shown (Lines 29-32 of revised manuscript with Track Changes).
Line 31: Correct ‘GAP’.
- We did correct ‘GAP’ as shown (Line 32 of revised manuscript with Track Changes).
Line 34: Villus height ‘to’ crypt depth.
- We did correct as shown (Line 36 of revised manuscript with Track Changes).
Line 36: Remove ‘for’.
- We did remove ‘for’ as shown (Line 37 of revised manuscript with Track Changes).
Line 90: Change ‘wide’ to ‘width’.
- We did change ‘wide’ to ‘width’ as shown (Line 123 of revised manuscript with Track Changes).
Line 146: Check for typo ‘≤ vs ≥’.
- Due to the typing error, thus, we did correct ‘≤ vs ≥’ as shown (Lines 183-184 of revised manuscript with Track Changes).
Line 158–160: Rephrase this sentence.
- We did improve the contents by modifying the sentences and make the new sentences as shown (Lines 195-200 of revised manuscript with Track Changes).
Line 162–163: Change ‘compared to’ to ‘compared with’.
- We did change ‘compared to’ to ‘compared with’ as shown (Lines 205-206 of revised manuscript with Track Changes).
Table 1: Check if ‘tryptophan’ would be better. Also, change fiber to ‘crude fiber’.
- We did change ‘trytophan’ to ‘tryptophan’ and ‘fiber’ to ‘crude fiber’ as shown in Table 1.
Line 236–238: Rephrase this sentence and provide a citation.
- We did improve the contents by modifying the sentences and make the new sentences and add the reference as shown (Lines 318-321 of revised manuscript with Track Changes).
Line 248–249: Change ‘term’ to ‘terms’.
- We did change ‘term’ to ‘terms’ as shown (Line 331 of revised manuscript with Track Changes).
Line 269–270: Rephrase this sentence. Consistent would be with findings/results.
- We did improve the contents by modifying the sentences and make the new sentences as shown (Lines 344-350 of revised manuscript with Track Changes).
Line 286–288: Check this sentence.
- We did agree with the reviewer. Thus, we did correct grammatical errors as shown (Lines 380-381 of revised manuscript with Track Changes).
Line 301–303: Why is red yeast an alternative source of antibiotics? Also, what does cholesterol do for antibiotic effect? This sentence would be an overstatement.
- Due to unclear the meaning of this sentence, thus, we did remove this sentence.
Line 312: Better to say ‘absorption of nutrients’. Substances could be nutritious or toxic.
- We did change ‘absorption of nutritious substances’ to ‘absorption of nutrients’ as shown (Line 408 of revised manuscript with Track Changes).
Line 314: Change ‘indicus’ to ‘indices’.
- We did change ‘indicus’ to ‘indices’ as shown (Line 410 of revised manuscript with Track Changes).
Line 318–320: Rephrase this sentence. Improved gut health is the outcome of other effects.
- We did improve the contents by modifying the sentences and make the new sentences and add the reference as shown (Lines 414-417 of revised manuscript with Track Changes).

Round 2
Reviewer 1 Report
This article is clear in thought, fluent in writing and properly revised, so I recommend that the article be published in this journal.
Author Response
January 15, 2022
Assistant Editor: Animals
Email: charlotte.shen@mdpi.com
Dear Editor of Animals and Reviewer 1,
Thank you for the review report of editor for manuscript (ID. Animals-19-1531926) included information that the manuscript needs to revise according to the comment of the reviewers. We have taken the useful comments of the referees and editor into account, and have made major revision as following page. We also describe all changes using the Track Changes and mostly agree with the comments. Thus, we hope our ability of the revised version will be satisfied by the reviewers and editor.
After the minor revision, we are fully aware of the revised submission and respectfully encourage you to give your utmost consideration to this revision manuscript. If you have any questions concerning with the manuscript, please feel free to contact me.
Sincerely yours,
Wanaporn Tapingkae, Ph.D
Corresponding author
e-mail: wanaporn.t@cmu.ac.th
In reply to review report of Assistant Editor dated January 11, 2022.
Subject: [Animals] Manuscript ID: animals-1531926 – Minor Revisions (Due Date: 15 January 2022)
We have taken the useful comments and queries of the referees and editor into account, and have made major revision as follows (answers are in the italic font).
Reviewer 1:
Comments and Suggestions for Authors:
This article is clear in thought, fluent in writing and properly revised, so I recommend that the article be published in this journal.
- Thank you very much for the review of our manuscript. We sincerely appreciate all valuable comments and suggestions, which helped us to improve the quality of the article.
Academic Editor Notes:
The plagiarism report indicated a high rate of similarity to other reports. It is strongly advised to rewrite material and methods section.
- We definitely agree with the Academic Editor. Thus, we have carefully revised the plagiarism according to the highlighted parts as shown, especially in the Material and Methods section (revised manuscript with Track Changes).
